# Game Theory-Based Energy-Efficient Clustering Algorithm for Wireless Sensor Networks

**DOI:** 10.3390/s22020478

**Published:** 2022-01-09

**Authors:** Xiao Yan, Cheng Huang, Jianyuan Gan, Xiaobei Wu

**Affiliations:** 1School of Automation, Nanjing University of Science and Technology, Nanjing 210094, China; xiaoyan8eli@njust.edu.cn (X.Y.); wuxb@njust.edu.cn (X.W.); 2School of Software Technology, Dalian University of Technology, Dalian 116024, China; jianyuangan@mail.dlut.edu.cn

**Keywords:** wireless sensor network, game theory, penalty mechanism, idle listening time, network lifetime

## Abstract

Energy efficiency is one of the critical challenges in wireless sensor networks (WSNs). WSNs collect and transmit data through sensor nodes. However, the energy carried by the sensor nodes is limited. The sensor nodes need to save energy as much as possible to prolong the network lifetime. This paper proposes a game theory-based energy-efficient clustering algorithm (GEC) for wireless sensor networks, where each sensor node is regarded as a player in the game. According to the length of idle listening time in the active state, the sensor node can adopt favorable strategies for itself, and then decide whether to sleep or not. In order to avoid the selfish behavior of sensor nodes, a penalty mechanism is introduced to force the sensor nodes to adopt cooperative strategies in future operations. The simulation results show that the use of game theory can effectively save the energy consumption of the sensor network and increase the amount of network data transmission, so as to achieve the purpose of prolonging the network lifetime.

## 1. Introduction

Wireless sensor networks are composed of a large number of sensor nodes deployed in a certain area, which provide feedback from information of the monitored area. As the connection between the physical environment and information environment, it is an important part of the Internet of things. At present, WSNs have been applied in the fields of ecological environment monitoring, military affairs, medical treatment, transportation, urban land use, and other fields [1,2,3]. Because of the wide application of WSNs, they have attracted the attention of many scholars in recent years. In WSNs, the sensor nodes perceive, transmit, and collect information in a cooperative way. This process will consume a certain amount of energy. However, the sensor nodes are powered by batteries, and the amount of power they carried is limited. Once the battery runs out, and the power supply cannot be replaced or supplemented in time, the transmission of monitoring information will be affected to a certain extent, and even the whole sensor network will be paralyzed [4,5]. Therefore, how to be as energy-efficient as possible in the case of limited energy, so as to extend the lifetime of the entire sensor network, has become a bottleneck problem in real applications of the sensor network.

In real applications, a battery with limited capacity is used to supply power for wireless sensor nodes, and the wireless sensor network is deployed in unattended outdoor or dangerous areas with a complex environment. The battery is used for power supply, and the monitoring area is uncontrollable, which is not conducive to energy supply or battery replacement of sensor nodes. To ensure the continuous work of WSNs, reducing the energy consumption of sensor nodes is the most effective way to prolong the lifetime of a sensor network [6]. Many experts and scholars have conducted several studies to optimize the energy consumption of sensor nodes to a certain extent [7,8,9,10,11]. After an extensive literature review, it was found that there has been little research on the energy consumption of sensor nodes in the idle listening stage. In fact, when a sensor node is in the active state, there will be substantial idle listening time, and the idle listening phase will also consume some energy [12,13]. If a large amount of energy is consumed at this stage, the energy used by the sensor node to collect and transmit information will be greatly reduced, thus shortening the lifetime of sensor nodes to a certain extent.

Moreover, the idle listening and sleep states of the sensor node in WSN have the characteristics of a game. Different strategies adopted by sensor nodes will bring different benefits and have different effects on the performance of sensor networks. On the one hand, sleeping of sensor nodes can minimize their energy consumption [14], but they will also consume a certain amount of energy when transitioning from the sleep state to the active state and from the active state to the sleep state [15]. If the energy consumed during state transition is too large, it will increase the energy consumption of the sensor nodes. On the other hand, the idle listening phase of sensor nodes will also consume a certain amount of energy. If the sensor nodes are idle listening for a long time, they will waste substantial unnecessary energy. In other words, the idle listening and the sleep states influence each other, which together determine the lifetime of the wireless sensor network.

To sum up, the main contributions of this work are as follows.

(1)Taking into account several aspects that affect the energy consumption of sensor nodes, a game model is established.(2)According to the energy consumption between the idle listening of sensor nodes and the transition of sensor nodes from the sleep state to the active state, the threshold value of sensor nodes entering the sleep state is determined.(3)In order to avoid the selfish behavior of sensor nodes when they go to sleep, a penalty mechanism is introduced to force the sensor nodes to adopt cooperative strategies in future operations. The optimal number of penalty rounds for sensor nodes with selfish behavior is proven.(4)The simulation results show that using the games to control the transition between the sleep state and active state of the sensor nodes can reduce their energy consumption, thereby effectively prolonging the lifetime of the network.

The remainder of the paper is organized as follows: Section 2 reviews related work; Section 3 describes the materials and methods; the simulation details and result analysis are provided in Section 4; a discussion and an overview of future work are provided in Section 5; the conclusion is introduced in Section 6.

## 2. Related Work

With the rapid development of the Internet of things, WSNs have been applied in various fields. The energy problem in WSNs has attracted more and more attention, and many experts and scholars have conducted several studies on energy efficiency. Under the condition of the limited energy of sensor nodes, how to maximize the lifetime of the network has become a hot topic in current discussion. In [7], they used two levels to optimize the formation of clusters and the choice of cluster heads (CHs), corresponding to two stages, i.e., the formation and operation of cluster. A new clustering optimization algorithm was proposed for wireless sensor networks with multilevel energy heterogeneity. The wireless sensor network based on this algorithm could balance the energy consumption of network nodes and prolong the life cycle of the network. In [8], three main aspects of WSNs were studied: accuracy, energy consumption, and computational complexity. A reverse asymmetric time synchronization framework was proposed for multi-hop wireless sensor networks with limited energy, and a beacon-free energy-saving time synchronization scheme based on reverse unidirectional message propagation was designed. On the premise of ensuring synchronization accuracy, the energy consumption and computational complexity were minimized. In [9], considering that the available environmental energy is unpredictable and changes with time, a method and its implementation to realize energy-neutral operation on energy collection wireless sensor nodes were proposed. The method utilizes adaptive duty cycle, which provides energy-neutral operation via an energy management circuit according to the energy available in the environment and the instantaneous energy state of the node. When there is no energy harvesting, the sensor node enters sleep mode to prolong its lifetime. In [10], the main purpose of the research was to design an energy embedded routing protocol based on an optimal update of the mobile sink. Furthermore, the mobile sink node is optimized to improve the network lifetime. In [11], a multi-hop clustering algorithm for wireless sensor networks was proposed to solve the problem of the low energy efficiency of nodes and long-term operation of an energy-harvesting wireless sensor network. Fuzzy logic was used to select the cluster head, so as to maintain the persistence of the network.

Game theory is a new branch of modern mathematics. It mainly considers the strategies and benefits for players, and it studies their optimization strategies. At present, the application of game theory in WSNs is becoming more and more extensive, mainly being used in energy efficiency, communication security, power control, data acquisition, etc. [16,17,18,19,20]. In recent years, the application of game theory in WSN energy efficiency has achieved particularly remarkable results. For example, in [21], a power control game model was established to optimize the balance between energy consumption and data packet transmission performance. Specifically, this paper first characterized the tradeoff using a multivariable optimization problem, with the goal of balancing the outage performance and the network lifetime. In [22], an energy-efficient clustering algorithm combined with game theory was proposed, using a dual-cluster-head mechanism to reduce the energy consumption of the sensor nodes. In [23], the existing distributed energy efficient clustering (DEEC) protocol for heterogeneous WSNs was improved. Game theory was used as an optimization algorithm, and the probability of the node becoming a cluster head was adjusted on the basis of residual energy, thus prolonging the lifetime of clusters. In [24], an energy-efficient clustering algorithm based on game theory was proposed. In the cluster head selection phase, each node competes as a potential cluster head by joining a localized clustering game, and a potential cluster head is selected to be a real cluster head through a properly designed probability method. In [25], a distributed clustering protocol for WSNs based on mixed game theory was proposed. Each node is modeled as a player, which can selfishly choose its own strategies to be or not be the cluster head. Furthermore, the payoff of each node was defined when choosing different strategies, in which the degree of nodes and distance to base station were both considered. In [15], a new approach was introduced by mixing a noncooperative game theory technique with a decentralized clustering algorithm. The method uses game theory to control the activities of a sensor node and its neighbors, so as to limit the number of forwarded messages and maximize the lifetime of the WSN. In [26], a cooperative game theory approach to energy-efficiency coverage in wireless sensor networks was put forward. The interaction between sensor nodes was modeled as a cooperative bargaining game. In this game, each sensor node can meet the needs of application awareness while minimizing energy consumption. In [27], a game theory method for balancing energy consumption in clustered wireless sensor networks was proposed. Considering the energy imbalance of sensor nodes, a penalty mechanism was introduced to force nodes with higher energy to actively become cluster heads. The clustering algorithm based on game theory proposed in this paper was proven to have a good energy balance performance. In [28], a better-distributed routing algorithm based on Q-learning game theory for underwater wireless sensor networks was introduced. Its Q-learning game paradigm captured the dynamics of the underwater sensor networks system in a decentralized and distributed manner. This method is effective in energy efficiency and can effectively extend the network lifetime. In [29], a heterogeneous game theoretical clustering algorithm called mobile clustering game theory 1 was proposed for energy optimization in a heterogeneous mobile sensor environment. Energy optimization was achieved through energy-efficient cluster head election and multipath routing in the network. In [30], a game theory- and enhanced ant colony-based mobile sink route selection and data gathering technique was presented, combining the optimal decision-making skill of game theory in selecting the best rendezvous points. GTAC-DG helped to reduce data transmission and management, energy consumption, and data transmission delay. In [31], a noncooperative game theory power control strategy based on CDMA WSNs was proposed. In this paper, the Nash equilibrium of power control strategy was discussed on the basis of noncooperative game theory. It could save sensor node energy, obtain good performance for the whole network, and greatly improve the network lifetime. In [32], an energy consumption optimal model of node cooperation based on game theory was proposed. In this model, the main energy consumption factor of the node-communication energy consumption was taken as the independent variable, and the payment function of the residual energy of the node was established. Moreover, it was verified that the optimal game model of the energy consumption of node cooperation had better stability.

The above research was optimized in terms of the clustering algorithm, power control, CH selection, and other aspects of sensor nodes, greatly improving the lifecycle of sensor networks. However, the energy consumption of sensor nodes during idle listening time was rarely considered. In fact, the energy consumption of this state will also have a great impact on the lifetime of the sensor node [12,13]. In this paper, we propose a game theory-based energy-efficient clustering algorithm (GEC), where we use game theory to discuss the transition of the sensor node between the active state and the sleep state, thereby reducing unnecessary energy consumption of sensor nodes in the idle listening stage, and avoiding the energy burden caused by frequent switching of the active state and the sleep state. This allows maximizing the benefits of sensor nodes and achieving the purpose of prolonging the network life cycle.

## 3. Materials and Methods

### 3.1. Study Object and WSN Deployment

In this paper, 100 sensor nodes with the same initial energy were randomly deployed in an area of 100×100, and the sink node was located in the center of the deployment area, i.e., (50, 50). Once the sensor node was deployed, its location did not move. Sensor nodes needed to continuously monitor the deployment area, and energy was no longer supplied.

### 3.2. Network Model

A WSN is a network randomly composed of a certain number of sensor nodes, which together form multiple clusters, and each cluster includes a cluster head node and multiple cluster members (CMs). Each CM determines whether it belongs to a cluster according to its corresponding location. Furthermore, each CM has a certain sensing range, within which each node can collect data from the monitored object and send the collected data to the corresponding CH, which then sends the data to the sink node. The network diagram used in this paper is shown in Figure 1.

When the sensor node is working, it is usually divided into two states: active state and sleep state. The sensor node in the active state can be divided into two stages: the working stages when the node needs to send and receive data, and the idle listening stages when there are no data to send and receive. When the sensor node is in sleep state, all functions are turned off to minimize energy consumption. In this state, the energy consumption of the sensor node can be considered as 0. In this paper, it is assumed that the sensor nodes are reasonable, while the energy is limited and distributed in harsh environments. For its own benefit, the sensor node should be as energy-efficient as possible to achieve the purpose of maximizing lifetime. Therefore, on the basis of ensuring network performance, when the sensor node is in the idle listening stage, it should enter a dormant state as much as possible, so as to reduce energy consumption and maximize the network lifetime.

Figure 2 shows the working schematic diagram of the sensor node, where Energy A–S represents the energy consumed by the sensor node in the transition from the active state to the sleep state, and Energy S–A represents the energy consumed by the sensor node in the transition from the sleep state to the active state.

### 3.3. Energy Model

The energy consumption of a sensor node is related to many factors. The main factors that directly influence the energy consumption of sensor nodes are discussed below. In order to calculate the residual energy of the sensor node, it is necessary to study various energy consumption factors of sensor nodes during the data processing. Therefore, the notations related to the energy consumption of sensor nodes are provided in Table 1.

#### 3.3.1. Energy Cost of Sensing

When the sensor works, it first senses the surrounding environment, and then acquires the sensing data from the environment [33]. In addition, the energy consumed by sensor nodes is related to the sensor itself. The sensing energy consumed by different sensors is different. Generally, the sensing energy consumption of sensor node $S_{i}$ is defined as
(1)$$E_{s}\left(S_{i}\right)=L\left(S_{i}\right)*I\left(S_{i}\right)*T\left(S_{i}\right)*V_{s},$$
where $I\left(S_{i}\right)$ is the needed amount of current, $V_{s}$ is the voltage supply, and $T\left(S_{i}\right)$ is the duration to detect and collect $L\left(S_{i}\right)$ bits of data.

#### 3.3.2. Energy Cost for Processing

When the sensor node reads and stores data, it also consumes energy [34]; thus, the energy consumption can be expressed as:(2)Ep(Si)=L(Si)∗Vs8(IWrite∗TWrite+IRead∗TRead),
where $T_{Write}$ and TRead indicate the durations taken by the sensor node in writing and reading $L\left(S_{i}\right)$ bits of data, and $I_{Write}$ and IRead denote the current required for the sensor node to write and read per bit of data.

#### 3.3.3. Energy Consumption for Communicating

The communication energy consumption of sensors mainly involves the energy consumption when sending and receiving information [15]. Hence, the energy consumed by the sensor node when sending information can be expressed as
(3)$$E_{t}\left(S_{i}\right)=\{\begin{array}{cc} L\left(S_{i}\right)*E_{elec}+L\left(S_{i}\right)*E_{fs}*d^{2} & \begin{array}{cc} when & d<d_{0} \end{array}\\ L\left(S_{i}\right)*E_{elec}+L\left(S_{i}\right)*E_{mp}*d^{4} & \begin{array}{cc} when & d>d_{0} \end{array} \end{array},$$
where $E_{elec}$ is the energy consumed to receive or transmit per bit message, whereas the constants $E_{fs}$ and $E_{mp}$ depend on the transmitter amplifier model, representing the free space model and the multipath model, respectively. d is the distance between two sensor nodes. $d_{0}$ is the threshold distance, which can be expressed as
(4)$$d_{0}=\sqrt{E_{fs}/E_{mp}}.$$

The energy consumed by the sensor node when receiving information can be expressed as
(5)$$E_{r}\left(S_{i}\right)=L\left(S_{i}\right)*E_{elec}.$$

#### 3.3.4. Energy Consumption in Transition from Sleep to Active Mode

When the sensor node is working, it will consume a certain amount of energy in the transition from the sleep state to the active state, and vice versa. However, the energy consumed for the switch from the active state to the sleep state is very small and, thus, negligible [15]. Therefore, the energy consumed by the sensor node to switch the working state is defined as
(6)$$E_{w}\left(S_{i}\right)=\frac{V_{s}}{2}*\left(I_{a}-I_{s}\right)*T_{as},$$
where $I_{a}$ is the current in the active state of the sensor node, $I_{s}$ is the current in the sleep state of the sensor node, and $T_{as}$ is the time required for the sensor node to switch from the sleep state to the active state.

#### 3.3.5. Total Energy Consumption for Sensor Node

According to Equations (1)–(6), the total energy consumption of the sensor node can be obtained as follows:(7)$$C_{i}=E_{s}\left(S_{i}\right)+E_{p}\left(S_{i}\right)+E_{t}\left(S_{i}\right)+E_{r}\left(S_{i}\right)+E_{w}\left(S_{i}\right).$$

### 3.4. Game Model

#### 3.4.1. Establishment of Game Model

In order to prolong the lifetime of a WSN, the sensor nodes can switch reasonably between the active state and the sleep state; thus, the dynamic switching of sensor nodes can be regarded as a game problem. The model can be expressed as follows:(8)$$GT=\left\{N,K,\{u_{i}\}\right\},$$
where N represents the player in the WSN. All sensor nodes in the network are involved in receiving and sending sensor information; hence, the game players include all sensor nodes in the network. In other words, the game player is sensor node $S_{i}$, i={1,2,⋯,n}. K is the strategic space of players. In view of the reasonable preference of sensor nodes, in this paper, the sensor node can determine the strategic space of game player by weighing the energy consumption, i.e., the sensor node enters the sleep state from the active state, and the sensor node does not enter the sleep state from the active state. $\left\{u_{i}\right\}$ represents the utility function of i player,
(9)$$u_{i}\left(s_{i},s_{-i}\right)=U_{i}\left(s_{i},s_{-i}\right)-C_{i}\left(s_{i},s_{-i}\right),$$
where $s_{i}$ is the strategy adopted by sensor node $S_{i}$, $s_{-i}$ is the strategy adopted by nodes other than sensor node $S_{i}$, $U_{i}\left(s_{i},s_{-i}\right)$ is the revenue function of sensor node $S_{i}$, and $C_{i}\left(s_{i},s_{-i}\right)$ is the cost function of sensor node $S_{i}$.

The wireless sensor network communicates data packet forwarding in a multi-hop manner. On the basis of the rational analysis of sensor nodes, the revenue function of a sensor node is defined as the reward obtained by a sensor node for successfully forwarding the data packet to the next sensor node. Therefore, the revenue function is defined as
(10)$$U\left(s_{i},s_{-i}\right)=G*P,$$
where G is the reward of a sensor node for successfully forwarding the data packet to the next sensor node, and P is the probability that the sensor node successfully forwards the data packet.

The sensor node has two working states, i.e., active state and sleep state. Therefore, $x_{j}\left(S_{i}\right)$ is defined as the two working states of sensor node $S_{i}$:(11)$$x_{j}\left(S_{i}\right)=\{\begin{array}{cc} 1 & Active\;\;\;\;\;\;mode\\ 0 & Sleeping\;\;\;\;mode \end{array}.$$

When the sensor node $S_{i}$ is in the sleep state, when it needs to forward data packets, it will adopt two different strategies, i.e., selfish and non-forwarding or cooperative and forwarding. Therefore, the two strategies adopted when the sensor node is in the dormant state are defined as
(12)$$y_{j}\left(S_{i}\right)=\{\begin{array}{cc} 1 & Forward\;\;\;data\;\;\;\;packet\;\\ 0 & Do\;\;\;not\;\;forward\;\;\;data\;\;\;\;packet\; \end{array}.$$

When the sensor node is working, the cost function is
(13)$$C_{i}\left(s_{i},s_{-i}\right)=\{\begin{array}{cc} E_{s}\left(S_{i}\right)+E_{p}\left(S_{i}\right)+E_{t}\left(S_{i}\right)+E_{r}\left(S_{i}\right) & x_{j}\left(S_{i}\right)=1\;\;\;and\;\;\;y_{j}\left(S_{i}\right)=1\\ E_{p}\left(S_{i}\right)+E_{t}\left(S_{i}\right)+E_{r}\left(S_{i}\right)+E_{w}\left(S_{i}\right) & x_{j}\left(S_{i}\right)=0\;\;\;and\;\;\;y_{j}\left(S_{i}\right)=1\\ 0 & x_{j}\left(S_{i}\right)=0\;\;\;and\;\;\;y_{j}\left(S_{i}\right)=0 \end{array}.$$

#### 3.4.2. Determination of Sleep State Threshold

When the sensor node in the wireless sensor network is working, how to determine the transition between the sleep state and the active state differs as a function of the energy consumption of the sensor node. At the same time, it also brings different benefits to the players. In other words, the energy consumption of the sensor nodes can be reduced by switching to a sleep state at an appropriate idle listening time, thus prolonging the lifetime of the sensor network. In order to obtain the threshold of sleep state transition of the sensor node, the following definitions are established:

**Definition** **1.**
*Let*

$\;R=\left(K_{s_{i}}^{j}\right)$

* be a strategic section adopted by the sensor node*

$\;S_{i}\;$

*in the wireless sensor network in the time period of *

j

*,*

ρ

*be the strategic section set of sensor nodes, and*

$\;U^{+}\;$

*be the utility set obtained by sensor nodes. The utility obtained by the sensor node*

$\;S_{i}\;$

*using the strategic section *

R

*is*

$\;u_{i}\;\left(s_{i},s_{-i}\right)\in U^{+}$

*, and the utility obtained by the sensor node*

$\;S_{i}\;$

*using a different strategic section*

 R 

*is*

$\;{u^{\prime }}_{i}\left(s_{i},s_{-i}\right)\in U^{+}$

*. If*

$\;{u^{\prime }}_{i}(s_{i},s_{-i})\;$

*does not exist, make*

$\;{u^{\prime }}_{i}\left(s_{i},s_{-i}\right)≻u_{i}\left(s_{i},s_{-i}\right)$

*; this shows that the strategy *

R

*adopted by the sensor node*

$\;S_{i}\;$

*is effective.*


**Definition** **2.**
*In the wireless sensor network, the players who make decisions can only improve utility, denoting that the players are rational.*


According to the personal rationality and strategic effectiveness of the players in the wireless sensor network, the following theorems can be drawn:

**Theorem** **1.**
*In the wireless sensor network, the sensor node is rational and the strategy is effective; thus, so the threshold of switching from an idle listening state to sleep state is *

$T^{\prime }\left(S_{i}\right)$

*, i.e.,*

(14)
$$T^{\prime }\left(S_{i}\right)=\frac{\left(I_{a}-I_{s}\right)*T_{as}}{2*L\left(s_{i}\right)*I\left(s_{i}\right)}.$$



**Proof** **of** **Theorem** **1.**In the wireless sensor network, the idle listening time of the sensor node will fluctuate; however, but for the energy efficiency of sensor nodes, they will not enter the sleep state during all idle listening periods when working. When the sensor node is in the sleep state, the energy consumed by the sensor node in idle listening must be greater than the energy consumed by the sensor node when switching working state, i.e.,$$L\left(S_{i}\right)*I\left(S_{i}\right)*T^{\prime \prime }\left(S_{i}\right)*V_{s}\geq \frac{V_{s}}{2}*\left(I_{a}-I_{s}\right)*T_{as}$$
(15)$$\Leftrightarrow T^{\prime \prime }\left(S_{i}\right)\geq \frac{\left(I_{a}-I_{s}\right)*T_{as}}{2*L\left(s_{i}\right)*I\left(s_{i}\right)},$$
where $T^{\prime \prime }\left(S_{i}\right)$ is the idle listening time of sensor node in the active state. The threshold of the sleep state $T^{\prime }\left(S_{i}\right)$ is the minimum idle listening time when the sensor node changes from the active state to the sleep state. □

The Nash equilibrium [16] is an important solution to the optimization problems in game theory. If each player $S_{i}$ in the game adopts a strategy to form a strategy combination $\left(D_{S_{1}}^{*},D_{S_{2}}^{*},\cdots ,D_{S_{n}}^{*}\right)$, the strategy $D_{S_{i}}^{*}$ of any player $S_{i}$ is the best strategy for the strategy combination $\left(D_{S_{1}}^{*},D_{S_{2}}^{*},\cdots D_{S_{i-1}}^{*},D_{S_{i+1}}^{*},\cdots ,D_{S_{n}}^{*}\right)$ of the remaining players in the game. In other words, the utility function $u_{i}\left(D_{S_{1}}^{*},D_{S_{2}}^{*},\cdots D_{S_{i-1}}^{*},D_{S_{i}}^{*},D_{S_{i+1}}^{*},\cdots ,D_{S_{n}}^{*}\right)\geq u_{i}\left(D_{S_{1}}^{*},D_{S_{2}}^{*},\cdots D_{S_{i-1}}^{*},D_{S_{i}},D_{S_{i+1}}^{*},\cdots ,D_{S_{n}}^{*}\right)$ applies for any $D_{S_{i}}\in K$; thus, $\left(D_{S_{1}}^{*},D_{S_{2}}^{*},\cdots ,D_{S_{n}}^{*}\right)$ is the Nash equilibrium of GT.

Theorem 1 gives the threshold for the sensor node to enter the sleep state. From the utility function, it can be seen that, in the process of a staged game, the sensor nodes gain more utility by adopting the sleep strategy. According to the self-interest of the sensor nodes, when the sensor nodes reach the sleep threshold, they will take selfish actions to save their own energy; thus, in a staged game, the sensor nodes adopt the sleep strategy as the Nash equilibrium of the game. However, when the sensor nodes are working, the game is a repetitive process. It is necessary to avoid sensor nodes using only a staged Nash equilibrium, as this will lead to the decline of the overall performance of the network. In this paper, a penalty mechanism is introduced to ensure the Nash equilibrium of the whole network.

#### 3.4.3. Penalty Mechanism of Sensor Node

In WSN, sensor nodes lack knowledge of the whole network information, and sensor nodes often adopt noncooperative strategies. In other words, sensor nodes can only choose the optimal strategy according to their own benefit, but they are not optimal from the standpoint of the entire network [35,36]. Due to the rational preference, once the sensor node enters the sleep state, it may choose to stay asleep and stop receiving and forwarding data packets to save energy consumption. This selfish behavior of node will lead to a delay or failure of data packet transmission, which will directly affect the operation mechanism of the whole wireless sensor network. In order to prevent the sensor node from taking selfish behavior, a penalty mechanism is introduced to encourage sensor nodes to wake up actively to send and receive data packets in a sleep state.

In order to ensure the normal operation of the wireless sensor network, the specific measures of the punishment mechanism adopted in this paper are as follows: when the sensor node in the wireless sensor network takes selfish behavior, the network will immediately mark the node and locate its ID. In the subsequent M rounds, the node is forced to remain in an idle listening state even if the idle listening time is longer than the sleep state threshold. The sensor nodes pay a certain price for their previous selfish behavior before, which also has a certain deterrent effect on the sensor node, forcing it to wake up in time to receive the forwarding data packets in a future sleep state.

In the condition of ensuring a sufficient deterrent to selfish sensor nodes, the premature death of sensor nodes due to too many penalty rounds can be avoided, thus destroying the balance of strategy. Therefore, how to determine the penalty rounds for sensor nodes is very important to prolong the lifetime of the whole wireless sensor network.

In order to better determine the penalty rounds of sensor nodes, this paper first defines the different benefits of the active state, sleep state, and penalty state of sensor nodes in the sensor network, thereby obtaining the utility matrix U of the sensor node as follows:(16)$$U=\left[\begin{array}{cc} u_{11} & u_{12}\\ u_{21} & u_{22}\\ u_{31} & u_{32} \end{array}\right].$$

In the matrix, $u_{\alpha \beta }$ represents the utility obtained after the sensor node takes an action, where α=1 means that the sensor node takes a wake up action, α=2 means that the sensor node takes a sleep action, and α=3 means that the sensor node is punished. β=1 means that the sensor node will forward the data packets to the next sensor node or sink node, and β=2 means that the sensor node will not forward the data packet to the next sensor node or sink node.

Since the sensor node is working, each data forwarding of the sensor node is a game process, and the sensor node will forward many times in its lifetime; therefore, we regard the forwarding process of the sensor nodes as a repeated game [37]. According to the principle of repeated game, in each game of the sensor node, the revenue obtained by the sensor node will have a discount factor δ (0<δ<1); if the value is larger, it means that the sensor node pays more attention to the future long-term gains. According to the discount criterion, the cumulative revenue function $U_{i}$ of the sensor node can be obtained as follows:(17)$$U_{i}=\sum_{m=1}^{M}\delta^{m-1}U_{im},$$
where $U_{im}$ is the revenue obtained by the sensor node $S_{i}$ in the m round.

It is assumed that the sensor node $S_{i}$ adopts a noncooperative strategy when working. In other words, when the idle listening time of the sensor node exceeds its sleep threshold after a packet is forwarded, the node will take selfish behavior after entering the sleep state and will no longer forward packet. Then, the network will mark the ID of the sensor node and force the node to stay awake in the next M rounds. According to the revenue function and cost function of the game player, it can be concluded that the utility obtained by the sensor node in the previous round and the subsequent penalty M rounds is as follows:(18)$$u_{22}+\sum_{m=1}^{M}{\left(u_{31}\right)}_{m}=0+\sum_{m=1}^{M}\delta^{m-1}U_{im}-\sum_{m=1}^{M}\left[E_{sm}\left(S_{i}\right)+E_{pm}\left(S_{i}\right)+E_{tm}\left(S_{i}\right)+E_{rm}\left(S_{i}\right)\right],$$
where ${\left(u_{31}\right)}_{m}$ represents the utility obtained by the sensor node $S_{i}$ when forwarding packets in the m round penalty state, $E_{sm}\left(S_{i}\right),E_{pm}\left(S_{i}\right),E_{tm}\left(S_{i}\right)$, and $E_{rm}\left(S_{i}\right)$ are the energy costs of sensor node $S_{i}$ for sensing, processing, sending, and receiving data, respectively, in the m round.

It is assumed that the sensor node $S_{i}$ adopts a cooperative strategy when working. In other words, the sensor node can always ensure the forwarding of the data packet, whether it is in the active state or the sleep state. To simplify the discussion, it is assumed that the sensor node will go to sleep when it reaches the threshold $T^{\prime }\left(S_{i}\right)$. Therefore, it can be concluded that the utility obtained by the node in the previous round and the subsequent M rounds is
(19)$$\begin{array}{l} u_{21}+\left(1-p\right)*\sum_{m=1}^{M}{\left(u_{11}\right)}_{m}+p*\sum_{m=1}^{M}{\left(u_{21}\right)}_{m}\\ =G*P-\left[E_{p}\left(S_{i}\right)+E_{t}\left(S_{i}\right)+E_{r}\left(S_{i}\right)+E_{w}\left(S_{i}\right)\right]\\ +\left(1-p\right)*[\left(\sum_{m=1}^{M}\delta^{m-1}U_{im}\right)-\sum_{m=1}^{M}\left(E_{sm}\left(S_{i}\right)+E_{pm}\left(S_{i}\right)+E_{tm}\left(S_{i}\right)+E_{rm}\left(S_{i}\right)\right)],\\ +p*\left[\left(\sum_{m=1}^{M}\delta^{m-1}U_{im}\right)-\sum_{m=1}^{M}\left(E_{pm}\left(S_{i}\right)+E_{tm}\left(S_{i}\right)+E_{rm}\left(S_{i}\right)\right)-M*E_{w}\left(S_{i}\right)\right] \end{array}$$
where ${\left(u_{11}\right)}_{m}$ and ${\left(u_{21}\right)}_{m}$ represent the utility of forwarding data packets of sensor nodes in the active state and sleep state, respectively; p is the proportion of the sensor node $S_{i}$ entering the sleep state in the M rounds. In this paper, we define the average distribution of actual idle listening time within the maximum idle listening time; therefore, the probability density function of idle listening time t is as follows:(20)$$f\left(t\right)=\{\begin{array}{cc} \frac{1}{T_{\max}} & t\in \left(0,T_{\max}\right)\\ 0 & Other \end{array},$$
where $T_{\max}$ is the maximum idle listening time of the sensor node in actual work. Therefore, it can be concluded that the probability of the sensor node $S_{i}$ entering the sleep state in the M round is as follows:(21)$$p=\int_{T^{\prime }\left(S_{i}\right)}^{T_{\max}\left(S_{i}\right)}\frac{1}{T_{\max}}dt=1-\frac{T^{\prime }\left(S_{i}\right)}{T_{\max}\left(S_{i}\right)}.$$

In order to ensure the balance of strategies and avoid the premature death of sensor nodes caused by excessive punishment, which can not only deter selfish nodes, but also prolong the lifetime of sensor nodes to the maximum, the following relationship is established:(22)$$u_{22}+\sum_{m=1}^{M}{\left(u_{31}\right)}_{m}\leq u_{21}+\left(1-p\right)*\sum_{m=1}^{M}{\left(u_{11}\right)}_{m}+p*\sum_{m=1}^{M}{\left(u_{21}\right)}_{m}.$$

Substituting Equations (18) and (19) into Equation (22), we can get
(23)$$p*M*E_{w}\left(S_{i}\right)\leq G*P-\left[E_{p}\left(S_{i}\right)+E_{t}\left(S_{i}\right)+E_{r}\left(S_{i}\right)+E_{w}\left(S_{i}\right)\right]+p*\sum_{m=1}^{M}E_{sm}\left(S_{i}\right).$$

Since the idle listening time is evenly distributed, the average idle listening time without entering the sleep state is $\frac{T^{\prime }\left(S_{i}\right)}{2}$; thus, the following relationship can be obtained:(24)$$\sum_{m=1}^{M}E_{sm}\left(S_{i}\right)\approx M*\frac{T^{\prime }\left(S_{i}\right)}{2}*L\left(S_{i}\right)*I\left(S_{i}\right)*V_{s}.$$

Substituting Equation (24) into Equation (23), we can get
(25)$$M\geq \frac{2*\left[E_{p}\left(S_{i}\right)+E_{t}\left(S_{i}\right)+E_{r}\left(S_{i}\right)+E_{w}\left(S_{i}\right)\right]-2*G*P}{p*\left[2*E_{w}\left(S_{i}\right)-T^{\prime }\left(S_{i}\right)*L\left(S_{i}\right)*I\left(S_{i}\right)*V_{s}\right]}.$$

Equation (25) gives the range of the number of rounds to punish the sensor node $S_{i}$ taking selfish behavior under the punishment mechanism. According to the definition of the Nash equilibrium, it can be determined that, when the number of penalty rounds M takes the minimum value, it is the Nash equilibrium of the whole network. In this case, it can not only ensure a sufficient deterrent effect on the sensor nodes, but also avoid premature energy depletion of the sensor node due to the excessive penalty rounds.

### 3.5. Algorithm Description

The sensor nodes are responsible for collecting and transmitting information in the wireless sensor network. In this paper, it is set that the sensor nodes can reasonably switch between idle listening and sleep states, thus reducing unnecessary energy consumption of sensor nodes during idle listening time, as well as avoiding the energy burden caused by frequent switching between idle listening and sleep states. Combining the game process of sensor nodes in idle listening and sleeping states, the steps of the proposed method for sensor nodes in the wireless sensor network are given in Algorithm 1.
**Algorithm 1** Proposed Algorithm1.Initialize:2.*N* = total nodes3.*Dead =* 0 //the number of dead nodes.4.Begin5.**for** *i* = 1:*N*6.   **if**
$S{\left(i\right)}_{\cdot }E>0\;\;$ //If node is alive7.    Cluster formation8.    Record the ID of node9.    **if** The sensor nodes need to forward the data10.       The sensor nodes forward the data11.    **else**12.       Calculate the sleep threshold $T^{\prime }\left(S_{i}\right)$13.       **if** $T\left(S_{i}\right)\leq T^{\prime }\left(S_{i}\right)$
14.        The sensor node remains idle listen15.       **else**
16.        The sensor node will enter the sleep state from idle listening17.       **end if**
18.    **end if**
19.   **else**
20.    *Dead = Dead* +121.    **if**
*Dead* ≥ *N*22.       End of simulation23.    **end if**
24.   **end if**
25.**end for**

The normal operation of a wireless sensor network is not only related to the number of dead sensor nodes, but also depends on the number of dormant nodes. In order to maintain the normal operation of the sensor network, a penalty mechanism is introduced to punish the sensor nodes in the sensor network for taking selfish behavior and force the sensor nodes to adopt cooperative strategies in future network operation. Therefore, according to the characteristics of the sensor network, the steps of the proposed method for the sensor network is given in Algorithm 2.
**Algorithm 2** Proposed Algorithm1.Initialize:2.*r* = current round3.Begin4.**for** *r* = 1:max5.   **if** the sensor network operating normally6.    The sensors nodes are clustered7.    **if** the sensor in the cluster adopt cooperative strategy8.       The sensor node decides their own state according to the game strategy9.    **else**
10.       Calculate the number of penalty rounds *M*11.       Mark the ID of the nodes and punish *M* rounds 12.    **end if**
13.   **else**
14.    End of simulation15.   **end if**
16.**end for**

## 4. Analysis

In order to verify the effectiveness of the algorithm for the lifetime of the sensor node, in this paper, MATLAB was used to simulate the low-energy adaptive clustering hierarchy (LEACH) [38,39,40,41,42], distributed energy efficient clustering (DEEC) [43,44,45,46], improved LEACH-centralized (LEACH-C) [47,48], localized game theoretical clustering algorithm (LGCA) [24,27], and our algorithm. The advantages of WSN sensor nodes in terms of lifetime, data transmission, and node survival state are then evaluated. In this paper, 100 sensor nodes were randomly distributed in an area of 100×100, and the sink node was set at (50, 50). Once the sensor nodes were arranged, energy was no longer supplied. The specific parameter settings during simulation are shown in Table 2.

Figure 3 shows the dead node comparison among LEACH, DEEC, LEACH-C, LGCA, and the game theory-based energy-efficient clustering algorithm (GEC) proposed in this article. The slope of the death node change curve of the GEC algorithm was obviously lower than that of LEACH, DEEC, LEACH-C, and LGCA. After the first node death in the GEC algorithm, the sensor nodes across a large area began to die. After 3500 rounds, the speed of node death was significantly reduced, and the lifetime of the network was better extended.

Figure 4 shows a histogram of the five algorithms in terms of the number of rounds after which the first node dies (FND), the tenth node dies (TND), half of the nodes die (HND), and the last node dies (LND) in the sensor network. LEACH, DEEC, LEACH-C, LGCA, and GEC exhibited FND after 1576, 1611, 1795, 1808, and 1888 rounds, respectively. Compared with LEACH, DEEC, LEACH-C, and LGCA, the number of FND rounds in GEC was extended by 19.80%, 17.19%, 5.18%, and 4.42% respectively. LEACH, DEEC, LEACH-C, LGCA, and GEC exhibited TND after 1770, 1841, 1995, 2089, and 2154 rounds, respectively. Compared with LEACH, DEEC, LEACH-C and LGCA, the number of TND rounds in GEC was extended by 21.69%, 17.00%, 7.97%, and 3.11%, respectively. LEACH, DEEC, LEACH-C, LGCA, and GEC exhibited HND after 2299, 2242, 2664, 2787, and 2855 rounds, respectively. Compared with LEACH, DEEC, LEACH-C, and LGCA, the number of HND rounds in GEC was extended by 24.18%, 27.34%, 7.17%, and 2.44% respectively. LEACH, DEEC, LEACH-C, LGCA, and GEC exhibited LND after 3417, 3379, 3817, 3952, and 4343 rounds, respectively. Compared with LEACH, DEEC, LEACH-C, and LGCA, the number of LND rounds in GEC was extended by 27.10%, 28.53%, 13.78%, and 9.89% respectively. In this paper, GEC had an advantage in terms of the emergence of FND, TND, HND, and LND in the sensor network, thus prolonging the network lifetime and better ensuring the normal operation of the network.

Figure 5 shows the amount of data transmitted by the sensor network in its lifetime, i.e., the amount of data received by the sink node. Compared with LEACH, DEEC LEACH-C, and LGCA, the data transmission capacity of the GEC was increased by 482.52%, 172.20%, 10.27%, and 5.71%, respectively. To address the problem of limited energy in a WSN, the GEC in this paper can not only prolong the service life of the network, but also greatly increase the data transmission volume of the network.

Figure 6 shows the change in residual energy of the sensor network with network rounds for the five different algorithms. It can be clearly seen from the figure that the GEC algorithm used in this paper was obviously better than the other four algorithms under the same number of rounds. In addition, the slope of the GEC algorithm was the smallest among the five curves, indicating that the energy consumption of the GEC algorithm was also the lowest. For example, as shown in Figure 7, compared with LEACH, DEEC, LEACH-C, and LGCA, the energy savings of GEC in the 1000th round were 12.33%, 12.36%, 4.14%, and 1.51% respectively. Compared with LEACH, DEEC, LEACH-C, and LGCA, GEC saved 29.52%, 29.46%, 9.01%, and 3.23% energy, respectively, in the 1500th round. After 2000 rounds, compared with LEACH, DEEC, LEACH-C, and LGCA, GEC could save 70.53%, 79.05%, 20.14%, and 6.43% energy, respectively. In addition, GEC could save 185.09%, 188.37%, 32.05%, and 10.27% energy, respectively, after 2500 rounds compared to LEACH, DEEC, LEACH-C, and LGCA.

In addition, the FND, TND, HND, and LND with variation in nodes are provided in Figure 8 in stacked form. When deploying a different number of sensor nodes in the same area, i.e., a different number of sensor nodes were distributed in a network cross-section of 100 × 100, the algorithm adopted in this paper was still superior to the other four algorithms. Figure 9 shows the stacked bar chart of FND, TND, HND, and LND with the same number of sensor nodes (n=100) distributed in different cross-sections. As the deployment cross-section increased, the performance of the sensor network gradually declined. This is because the range of the deployment cross-section increased, and the distance for sensor nodes to transmit information increased; hence, the energy consumed by the sensor node to send information increased, while the lifetime of the sensor network decreased. However, under the same situation, the performance of this algorithm was better than the other four algorithms.

Figure 10 shows the change in network throughput when different numbers of sensor nodes were deployed in the same network cross-section (100 × 100). With the increase in the number of deployed sensor nodes, i.e., an increase in throughput, the GEC algorithm adopted in this paper always outperformed the other four algorithms. Figure 11 shows the change in network throughput when the same sensor nodes (n=100) were deployed in different network cross-sections. With the increase in network cross-section, the network throughput decreased, but the GEC algorithm adopted in this paper was always better than the existing protocols.

## 5. Discussion

In the same area, the same number of sensor nodes was deployed. Compared with the existing LEACH, DEEC, LEACH-C, and LGCA algorithms, the GEC algorithm adopted in this paper could effectively prolong the life cycle of the network (Section 4, Figure 3) and greatly improve the stability of the network (Section 4, Figure 4). The data throughput of the network was also increased (Section 4, Figure 5), and the energy of the sensor network could be effectively saved (Section 4, Figure 6 and Figure 7). When different numbers of sensor nodes were deployed in the same network cross-section, the GEC algorithm could still effectively improve the stability of the network and greatly increase the data throughput of the network (Section 4, Figure 8 and Figure 10). Similarly, when the same number of sensor nodes was deployed in different network cross-sections, the performance of the GEC algorithm in WSN was still better than the other four algorithms (Section 4, Figure 9 and Figure 11). This is because the sensor nodes could rationally enter the sleep state during idle listening time by using game theory, thereby reducing the energy consumption of sensor nodes and effectively improving the WSN performance.

The development of the Internet of things (IoT) has already had a great impact on people’s lives [49]. The IoT consists of three parts, sensor network, transmission network, and application network, and the sensor network composed of wireless sensor nodes is an important part of the IoT. Its function is to sense, collect, process, and transmit the information in the area covered by the network in a cooperative way, relaying the information to the users. Therefore, the sensor network plays a vital role in the development and application of the IoT. Because of the low cost, low power consumption, and fault tolerance of the wireless sensor network, even if some nodes are damaged, the whole network will not be paralyzed. This ensures that the sensor network can still provide accurate and reliable information in a harsh environment. Therefore, WSNs can be widely used in many areas such as the military field, forest environment monitoring, and intelligent transportation.

Energy is the basis of a node working, and the energy saving problem is an important factor that limits the development of WSNs and the IoT. Therefore, this paper proposed a game theory-based energy-efficient clustering algorithm, which can effectively reduce the energy consumption of sensor nodes in idle listening time and greatly improve the performance of the WSN. In future work, we will try to build a dynamic game model in a heterogeneous environment, so that the game model can be more widely used to improve the lifetime of WSNs and provide an energy guarantee for the application and development of WSNs and IoT.

## 6. Conclusions

In the condition of limited energy, in order to prolong the service life of sensor nodes in a WSN, a model based on repeated games was established, and the strategy adopted by sensor nodes was determined by studying the length of the idle listening stage. When the idle listening time is longer than the sleep threshold, the sensor node enters the sleep state to save energy consumption. When the idle listening time is less than the threshold of the sleep state, the sensor node remains in an idle listening state, which reduces the energy consumption caused by the transition between the sleep state and the active state of the sensor node. In order to prevent the sensor node from adopting a noncooperative strategy after going to sleep, this paper also proposed a punishment mechanism for the sensor node. This not only ensures strategic balance, but also acts as a deterrent to selfish nodes, forcing them to adopt cooperative strategies in future operations. The simulation results show that the game theory-based energy-efficient clustering algorithm is effective in saving energy consumption of the sensor network, increasing the data transmission of sensor nodes, and extending the lifetime of sensor nodes.

## Figures and Tables

**Figure 1 sensors-22-00478-f001:**
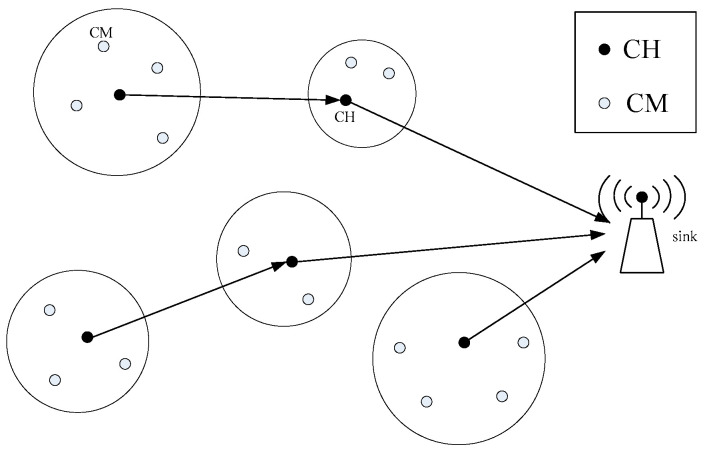
Schematic diagram of clustering algorithm.

**Figure 2 sensors-22-00478-f002:**
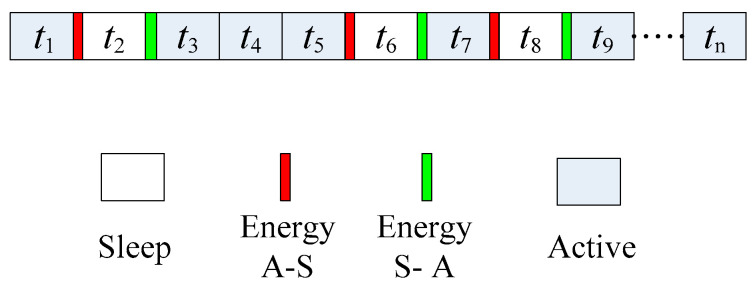
Sensor node scheduling table.

**Figure 3 sensors-22-00478-f003:**
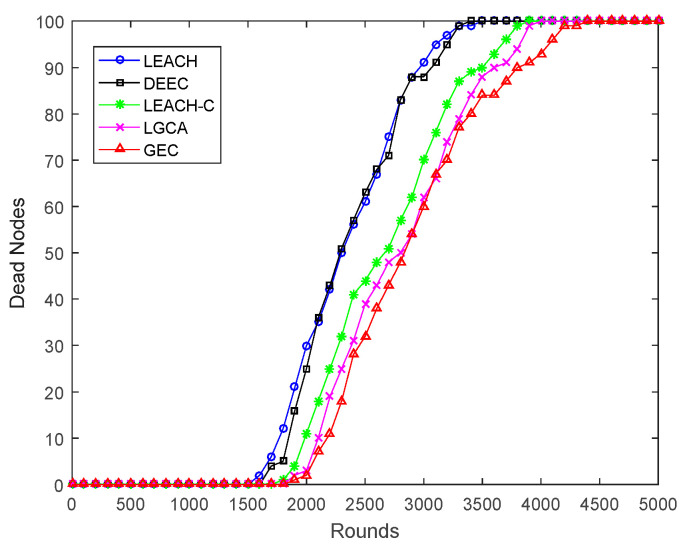
The network dead node comparison.

**Figure 4 sensors-22-00478-f004:**
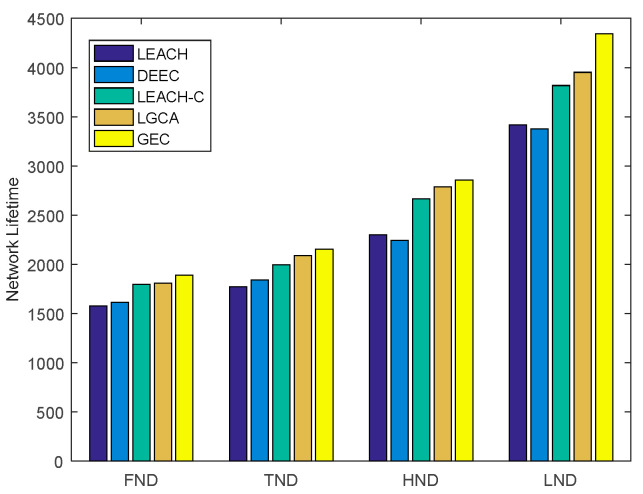
The network lifetime comparison.

**Figure 5 sensors-22-00478-f005:**
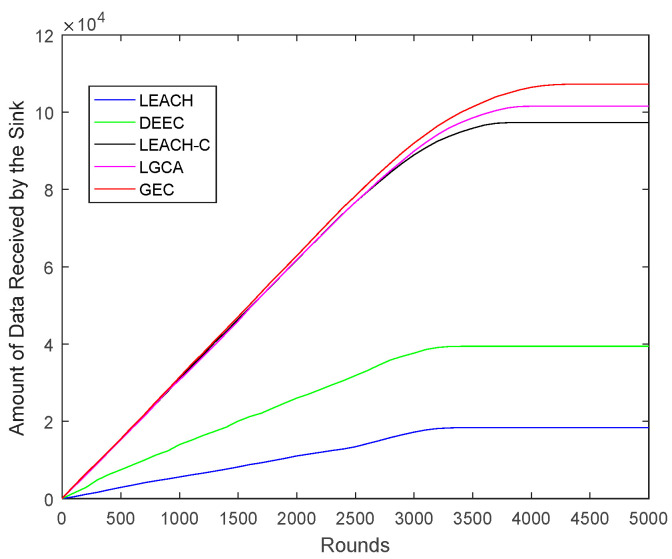
The network data transmission comparison.

**Figure 6 sensors-22-00478-f006:**
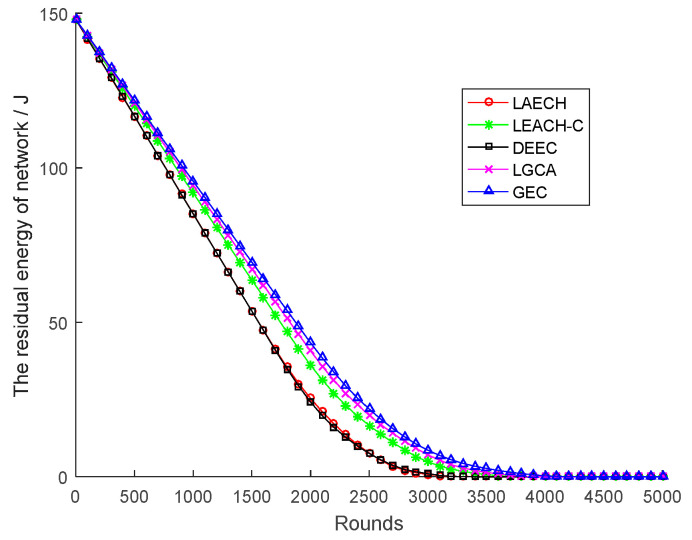
The residual energy of network.

**Figure 7 sensors-22-00478-f007:**
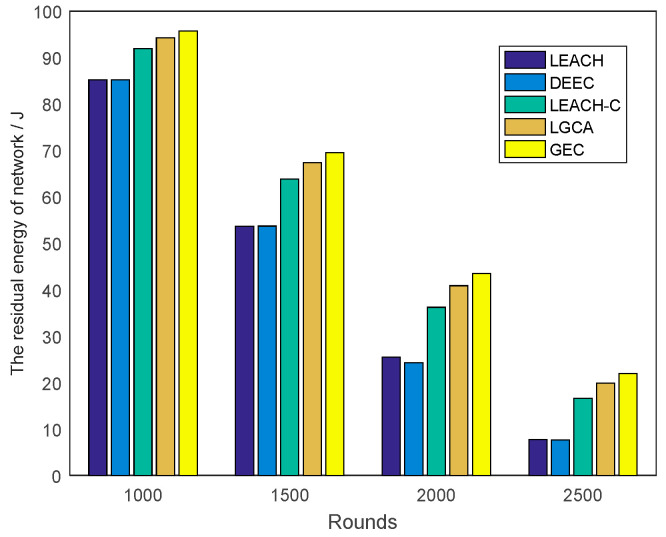
Comparison of residual energy of the network running different rounds.

**Figure 8 sensors-22-00478-f008:**
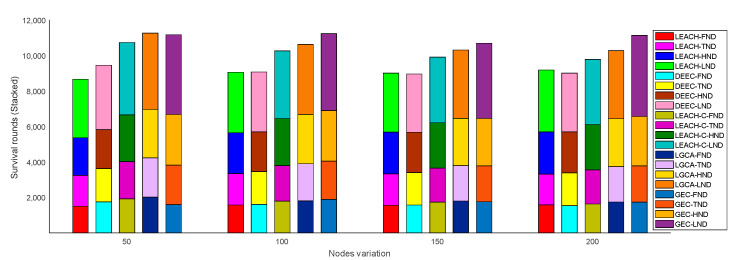
Network lifetime vs. node variation.

**Figure 9 sensors-22-00478-f009:**
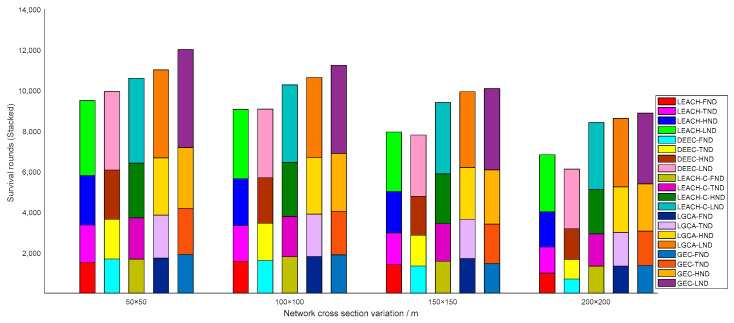
Network lifetime vs. network cross-section variation.

**Figure 10 sensors-22-00478-f010:**
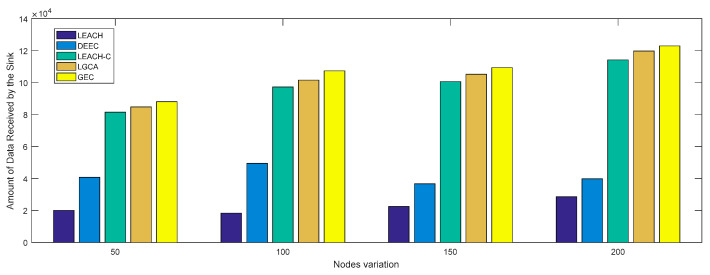
Throughput vs. node variation.

**Figure 11 sensors-22-00478-f011:**
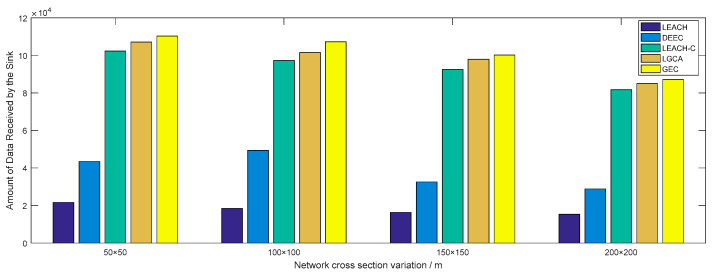
Throughput vs. network cross-section variation.

**Table 1 sensors-22-00478-t001:** Notations of parameters and variables.

Symbol	Description
n	Number of sensor nodes in the network
$S_{i}$	Sensor node where i={1,2,⋯,n}
$E_{s}\left(S_{i}\right)$	Energy cost for sensing
$E_{p}\left(S_{i}\right)$	Energy cost for processing
$E_{t}\left(S_{i}\right)$	Energy cost for sending
$E_{r}\left(S_{i}\right)$	Energy cost for receiving
$E_{w}\left(S_{i}\right)$	Energy consumption for sleep to active mode
$C_{i}$	Total energy consumption for sensor node

**Table 2 sensors-22-00478-t002:** Simulation parameters.

Parameters	Value
$\mathrm{Initial}\;\mathrm{energy}\;(J)\;E_{initial}$	1.0
$\mathrm{Number}\;\mathrm{of}\;\mathrm{iterations}\;r_{\max}$	5000
Size of date packet (bits)	4000
Proper percentage of CH nodes (%)	5
$\mathrm{Parameters}\;\mathrm{of}\;\mathrm{amplifier}\;\mathrm{energy}\;\mathrm{consumption}\;(\mathrm{pJ}/\mathrm{bit}/m^{4}$ $)\;E_{mp}$	0.0013
$\mathrm{Parameters}\;\mathrm{of}\;\mathrm{amplifier}\;\mathrm{energy}\;\mathrm{consumption}\;(\mathrm{pJ}/\mathrm{bit}/m^{2}$ $)\;E_{fs}$	10
$E_{elec}$ (nJ/bit)	50

## Data Availability

The study did not report any data.

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
