# Peer review of "Game Theory-Based Energy-Efficient Clustering Algorithm for Wireless Sensor Networks"

_sensors, 2022, doi:10.3390/s22020478_

Round 1
Reviewer 1 Report
The article is interesting.
I think that 8 sections is too much for an article of this side. The authors must condense some of the sections.
A materials and methods section is mandatory, as well as a discussion section. Those are major requirements in any article.
The authors must explain properly in the introduction why the subject is important and how the subject will be of major interest for the readers, as well as in the impact that this can have for real applications.
Reviewer 2 Report
The paper proposes a game theory based clustering approach for WSNs, in order to improve energy efficiency. While the paper is rigorous in it's approach, it needs to address the following concerns:
- While the paper presents a Nash equilibrium based non-cooperative game theoretic approach to improving lifetime of WSNs, similar approaches exist in the literature, as already outlined in the Related Work. The paper should highlight clearly the key contributions of the proposed approach in comparison to similar approaches.
- Further, the evaluation results compare the proposed algorithm with traditional non-game theoretic approaches such as LEACH. The paper will benefit from a comparison with other game theoretic approaches in the literature.
- It is claimed that one of the contributions is to force nodes to adopt cooperative strategies. The paper should highlight the key motivations for adopting a non-cooperative approach as opposed to cooperative ones, given that such techniques have been proposed in the literature to achieve similar outcomes, such as below:
- Banerjee, A., Gauthier, V., Labiod, H. and Afifi, H., 2013. Cooperation optimized design for information dissemination in vehicular networks using evolutionary game theory. arXiv preprint arXiv:1301.1268.
- While the proposed technique is for WSNs, the paper will benefit from a discussion on how the proposed technique can be applied to more realistic usecases and applications, such as IoT deployments.
Round 2
Reviewer 1 Report
The authors have addressed all my questions and for that reason I recommend the article to be published.
Reviewer 2 Report
The authors have addressed all the reviewer comments sufficiently and can be accepted.
This manuscript is a resubmission of an earlier submission. The following is a list of the peer review reports and author responses from that submission.
Round 1
Reviewer 1 Report
I think the paper is very good overall. The related literature is described in detail, the model and approach are carefully applied, and a comprehensive set of experiments are performed that establish improvement over the best prior approaches.
I suggest an extensive English proofreading by a native speaker.
There is a lot of notation that is hard to follow, but maybe that is inevitable for this paper.
Line 351: its should be it's
Line 379-381: this is the first time where Nash equilibrium is introduced. It should be described more clearly why this threshold constitutes a Nash equilibrium of the repeated game model. It isn't really clear to me where Nash equilibrium/game-theoretic analysis comes in the derivation as opposed to just single-agent reasoning.
For Figure 9, the caption should be on the same page as the figure.
Reviewer 2 Report
From the get-go, the paper is faulty with its language. The first sentence in the abstract starts with,
"Energy-efficient is one of the critical challenges ..." but it should be rather, "Energy-efficiency"
like, "Energy-efficiencey is one of the critical challenges ..."
The starting lines are also bland. There are hundreds of papers in this area in journals and conferences that have similar wordings! Then, there are other serious issues about the paper. Let me point out those here:
1. The paper is clearly untimely. The authors have not at all done a proper literature review. While some papers are linked, I find that the authors have missed some critical research papers in this area. For instance,
Thandapani, P., et a., "An energy-efficient clustering and multipath routing for mobile wireless sensor network using game theory," Int. Journal of Comm. Sys., Jan 2020, doi:10.1002/dac.4336
Likewise, other recent works are completely ignored. It seems that the paper was prepared quite some time ago. Hence, it lacks the professional merit required. In fact, some papers brought into the discussion have not even got the necessary emphasis but those are simple mention of the approaches without making those really relevant to this current work or about their comparative studies.
2. Game theory has already been explored and applied in this domain and there is nothing novel in this work. The key findings as claimed "the use of game theory can effectively save the energy consumption of the sensor network and increase the amount of network data transmission" are not at all new! The authors have ignored a large part of the previous works and hence, such technically dwarf work has been produced.
3. From time to time, the language issue makes the paper often awkward for reading. The paper must go through a language correction.
4. As for the technical side, I find very low novelty for the energy model or game model used.
5. The algorithm in Section 6 is presented in a very unprofessional way. Same for other such elements. The authors should look into the format how to present algorithm in proper way.
6. As the authors claim to do the performance analysis in MATLAB, they must have the associated codes for compared protocols. Perhaps, a GitHub link or personalized link would convince a reviewer about the other codes used for the protocols mentiond. The results otherwise is unclear. Also, the relative gain in exchange of the negative impact is not clear. What is the trade-off here? For instance, when you compare with basic LEACH or similar protocol?
7. The end-goal is basically energy-efficiency and thus, lifetime extension or prolonging. There are several other works with concrete experimentations on these issues. For instance,
Kalpana Murugan and Al-Sakib Khan Pathan, “Prolonging the Lifetime of Wireless Sensor Networks using Secondary Sink Nodes,” Telecommunication Systems, Volume 62, Issue 2, June 2016, Springer US, pp. 347-361, DOI: 10.1007/s11235-015-0079-5.
Ramadhani Sinde et al., "Lifetime improved WSN using enhanced-LEACH and angle sector-based energy-aware TDMA scheduling," Cogent Engineering, Volume 7, 2020, Issue 1, doi:10.1080/23311916.2020.1795049
A list of papers is mentioned here:
Felicia Engmann et al., "Prolonging the Lifetime of Wireless Sensor Networks: A Review of Current Techniques," Wireless Comm and Mob. Computing, Volume 2018 |Article ID 8035065 | doi:10.1155/2018/8035065
There are various ways of prolonging the lifetime but clustering in general has been extensively studied. Even if other issues are brought in, the authors should focus on showing concrete evidence of lifetime increase as the proof in this paper is insufficient.
Overall, given the timeliness, novelty, overall technical merit, the paper is not up to the mark for this venue. My recommendation is to reject it.